# Revealing the Meissner Corpuscles in Human Glabrous Skin Using In Vivo Non-Invasive Imaging Techniques

**DOI:** 10.3390/ijms24087121

**Published:** 2023-04-12

**Authors:** Victor Hugo Pacagnelli Infante, Roland Bennewitz, Anna Lena Klein, Martina C. Meinke

**Affiliations:** 1INM-Leibniz Institute for New Materials, 66123 Saarbrücken, Germany; 2Center of Experimental and Applied Cutaneous Physiology (CCP), Department of Dermatology, Venereology and Allergology, Charité-Universitätsmedizin Berlin, Corporate Member of Freie Universität Berlin and Humboldt Universität zu Berlin, Charitéplatz 1, 10117 Berlin, Germany; 3Department of Physics, Saarland University, 66123 Saarbrücken, Germany

**Keywords:** Meissner corpuscles, glabrous skin, laser scan microscopy, optical coherence tomography, tactile perception

## Abstract

The presence of mechanoreceptors in glabrous skin allows humans to discriminate textures by touch. The amount and distribution of these receptors defines our tactile sensitivity and can be affected by diseases such as diabetes, HIV-related pathologies, and hereditary neuropathies. The quantification of mechanoreceptors as clinical markers by biopsy is an invasive method of diagnosis. We report the localization and quantification of Meissner corpuscles in glabrous skin using in vivo, non-invasive optical microscopy techniques. Our approach is supported by the discovery of epidermal protrusions which are co-localized with Meissner corpuscles. Index fingers, small fingers, and tenar palm regions of ten participants were imaged by optical coherence tomography (OCT) and laser scan microscopy (LSM) to determine the thickness of the stratum corneum and epidermis and to count the Meissner corpuscles. We discovered that regions containing Meissner corpuscles could be easily identified by LSM with an enhanced optical reflectance above the corpuscles, caused by a protrusion of the strongly reflecting epidermis into the stratum corneum with its weak reflectance. We suggest that this local morphology above Meissner corpuscles has a function in tactile perception.

## 1. Introduction

Touch and tactile perception are central in our communication and well-being. Throughout our entire life, we explore the world with our senses [1]. Löken and Olausson (2010) [2] have described the skin as a social organ, an interface between the external world and our own body. This interface is crucial for how we feel and interact with one another. The skin of our hands plays an important role in the haptic perception, since our hands are specialized to touch and explore surfaces and objects [3,4,5,6]. Nevertheless, physiological studies focusing on the hand’s skin are scarce compared to those on hairy skin. This imbalance is a barrier to developing a better comprehension of possible correlations between tactile perception and skin pathophysiology.

The skin of our hands is also known as glabrous skin, without hairs, with a thick stratum corneum (SC) compared to hairy skin, and with a high density of sweat glands. Glabrous skin is also found on the lips and soles of humans. Its innervation by specialized nerves makes glabrous skin an important link between touch and perception [7,8,9,10]. These nerves lead to the perception of subtle tactile details and are localized in the basal layer of the epidermis (Merkel cells), the dermal–epidermal junction (DEJ) (Merkel cells, Meissner corpuscles) or the dermis (Pacinian corpuscles and Ruffini organs) [10,11]. 

Meissner corpuscles (MCs) are prominent among the above-mentioned mechanoreceptors. Their position in the DEJ and their neural response makes them sensitive to movement across the skin, and provides them with a key role in the active tactile exploration of surfaces and the perception of textures [8]. The basic anatomy of the MCs has been revealed by biopsies and animal models [11]. The connections between MC physiology, touch-related behavior, and tactile perception are the subjects of ongoing research [12].

It is also important to remember the clinical implications of MCs, such as the distribution reduction in diabetes mellitus patients [13], hereditary neurological disorders [14] or HIV neuropathy [15]. Most clinical studies focused on intraepithelial nerve fibers, while few investigated how the number or quality of corpuscles in human skin changes with pathologies of the peripheral or central nervous system [14,15]. Variations in the amount, physiology, and distribution of MCs can potentially serve as markers for disorders or patients with healing problems from diabetic neuropathy, as described by Garcia-Mesa et al. [16]. However, biopsies are an invasive and uncomfortable method to quantify mechanoreceptors in healthy participants in fundamental physiology studies.

Non-invasive optical techniques allow for a more suitable implementation of human skin physiology studies, by avoiding biopsies [17]. Only few studies have looked at the skin physiology of glabrous skin using non-invasive techniques, despite the necessity to understand the role of healthy skin physiology in tactile perception and how some diseases may affect the skin innervation [15,18]. Established in vivo microscopy techniques, such as optical coherence tomography (OCT) and laser scan microscopy (LSM) share the problem of the thick SC of glabrous skin limiting the optical access to the viable epidermis and the dermis, reducing the spatial resolution and affecting the quantification of MCs. An improved understanding of the skin morphology associated with MCs and its imaging by these noninvasive techniques is essential to introduce them as alternatives to biopsies in the quantification of MCs in glabrous skin.

In this study, we introduce protrusions of the epidermis into the stratum corneum above MCs as a feature of high salience in LSM images, which helps to localize these mechanoreceptors in image analysis. We compare OCT and LSM as in vivo, non-invasive, skin imaging techniques to choose the best method for counting MCs in the glabrous skin of the fingers and palm. The protrusions do not only guide the medical practitioner in the localization of MCs, but are also a phenomenon with impact on tactile perception and potential physiological importance in the future comprehension and management of skin diseases. 

## 2. Results

### 2.1. Comparing OCT and LSM as Non-Invasive In Vivo Techniques on Glabrous Skin

In the present study, two techniques were used with different spatial resolutions and possible applications (Figure 1). OCT provides the SC thickness (Figure 1A,B, blue arrow), the sweat gland morphology (Figure 1A,B, orange arrow), and allows for the visualization of the papillae, depending on the SC thickness (Figure 1B, gray arrows). The resolution of OCT is limited (axial resolution between 5 and 15 µm), and details such as the position and shape of MCs are missed. A very useful feature of this method is the imaging of longitudinal cross-sections and the fast quantification of SC thickness. 

LSM provides images of the MCs with a satisfactory resolution (axial resolution between 1 and 10 µm), despite the thickness of the SC in glabrous skin (Figure 1D, blue circles). The method also provides good images of sweat glands (Figure 1C orange arrows) and corneocytes (Figure 1C yellow contour).

In LSM, the laser wavelength can be chosen to improve the resolution of the different layers. The 488 nm laser provides better resolution in images of the upper layers if compared with the 756 nm wavelength (Figure 1E,G). The larger penetration depth of the 756 nm laser results in more details in images of deeper layers, which is particularly important for the study of MCs (Figure 1F,H).

### 2.2. LSM of Meissner Corpuscles

We observed the MCs in the dermal–epidermal junction (DEJ) of glabrous skin by LSM, presenting an ellipsoidal or spherical shape (Figure 2A). The appearance of MCs and their distinction from sweat glands is further demonstrated in Appendix A. They occur as singles, pairs, or triplets in the same papilla. The MCs have an interesting proximity to blood capillaries in the dermis (Figure 2B, blue arrow). The Appendix A provides additional images of MC morphologies and blood capillaries in the dermis (Appendix A). For a better representation of the blood capillaries in the dermal–epidermal junction, VivaStack images should be recorded with a 1.5 µm height increment. Overview imaging confirms that the MCs are most often situated next to the ridges (Figure 2C–E).

### 2.3. Individual Results for Participants 

The thickness of the SC and of the epidermis for each participant are presented in Table 1. The SC thickness was measured by OCT, the other features were determined by analysis of LSM images. No significant correlations of these quantities were found with the age or sex of participants.

### 2.4. Meissner Corpuscles Locally Alter the Morphology of the Epidermis and Stratum Corneum

In LSM VivaStack^®^ images recorded at an increasing depth, the transition from stratum corneum to the epidermis is indicated by a strong increase in reflectance, which is caused by the higher water content and other molecules of the epidermis. We discovered that this transition is locally shifted upward, above the location of the MCs. An example for a small finger is given in Figure 3. It is also possible to observe that the presence of MCs can increase the reflectance even more, which can be related to the aggregation of cells in the same portion.

At a depth of between 100 and 300 µm, according to the anatomic position and/or the participant, some regions show a much higher reflectance, which indicates the transition from the SC to the epidermis (Figure 3B and Appendix A). Images recorded between the SC and the epidermis reveal that MCs are found only in the regions which exhibited higher reflectance.

The upward shift of high reflectance was always found above MCs in comparison to adjacent regions without MCs. We interpret this upward shift of reflectance as a protrusion of the epidermis into the stratum corneum above MCs. The extent of this protrusion can be quantified by plotting the brightness of the LSM images (gray level index between 0 and 255) in the respective regions as a function of depth. One example is given in Figure 4A, where the average intensity level is plotted as a function of the depth for the small finger of all participants. Note that the depth is taken as a percentage of the thickness of each participant’s stratum corneum. Above MCs, the reflectance increases at the top of the protrusion at a depth which corresponds to 100% of the SC. In adjacent regions without MCs, the reflectance increases only at the dermal–epidermal junction, at a depth corresponding to 118% of the stratum corneum. The protrusions were more prominent with an average of 25 µm in the finger skin compared to an average of 12 µm in the palmar area (Figure 4B; *p* = 0.003). 

We also extracted the thickness of the epidermis from the LSM VivaStack^®^ results as the distance between the onset of epidermis and the DEJ. The epidermis was found to be thinner in the regions with MCs (71 ± 14 µm) if compared with regions without MCs (85 ± 17 µm) (Figure 4C; *p* < 0.001).

The height of the protrusions is positively correlated (R^2^ = 0.49; *p* < 0.001) with the SC thickness (Figure 5).

Our LSM results for glabrous skin show that the MCs are co-localized with protrusions of the epidermis into the stratum corneum and with a local reduction of the epidermal thickness. These findings are summarized in the sketch in Figure 6.

## 3. Discussion

So far, few studies have applied in vivo non-invasive skin imaging techniques to the study of Meisner corpuscles (MCs) in humans [13,15,18]. Furthermore, only selected anatomical regions have been investigated, with the small finger most explored because the optical access is better through its thinner stratum corneum (SC) and the small finger has been found to exhibit a similar density of MCs as the index finger [15,18]. We suggest that more application of non-invasive optical methods to the study of MCs is desirable, since variations in their number and distribution can be related to different diseases.

A review of available literature indicates that the LSM is the in vivo technique most often applied to the study of MCs [13,15,18,19]. The application of OCT was reported only for studies where the resolution of this technique was either improved, where a high axial resolution was not required. Libe et al. (2017) [20] introduced speckle-modulating OCT (SM-OCT), whereby the elimination of speckle noise originating from the sample provides clear images of MCs in the human finger pad skin. In the study reported here, standard OCT did not provide us with images of MCs (Figure 1A,B), but was very efficient in the quantification of other features, such as the SC thickness, in agreement with previous reports [21,22,23,24]. At present, the availability of LSM is limited, but indispensable for such studies, if biopsies are avoided.

Multiphoton tomography (MPT) allows for the non-invasive microscopy of the skin physiology, with a high spatial resolution [23,25]. In vivo studies of the human glabrous skin were not reported yet for MPT. One challenge may be the relatively thick SC in glabrous skin, which does not foreclose the high-resolution imaging of details in upper skin layers, such as sweat glands and corneocytes. 

In our LSM experiments, we successfully acquired images of glabrous skin up to 300 µm from its surface. For many participants, we could thus image MCs even in the index finger with its thickest stratum corneum. This resolution of LSM allowed us to describe the localization and morphology of MCs and some differences between the fingertips and the palm (Figure 2). The observation of proximity between MCs and blood capillaries in Figure 2D,E deserves further investigation in future studies of MC functionality. 

The concentration of MCs next to skin ridges suggests an enhancement of their function by the structure. Gerling and Thomas (2007) [26] hypothesized that the receptors for detecting shear stresses (MCs) reside at the tips of ridges, indicating a functional importance in the detection of stress. Ridges increase the surface area and, consequently, amplify slip pulses or vibrations that are to be detected by MCs (Figure 3B). Skedung et al. (2018) [19] acquired LSM images of the index finger and correlated the density of MCs with tactile perception, observing a reduction in elderly participants. This is the only study available which applies in vivo LSM to relate skin physiology to tactile perception.

The most important observation in this study is the protrusion formed above MCs, which increases the surface area between the epidermis and the SC. Knowing this feature can guide the identification of MCs by the LSM technique, even if a thick SC interferes with a clear imaging of MCs. Local enhancement of the reflectance at the junction between the SC and epidermis may serve as proxy for the direct imaging of MCs, and indicate the distribution of MCs for participants with a thicker SC. The main limitation of applying such a proxy is the uncertainty of the number of MCs per papillae that may vary from one to three (Figure 2A–D). In some images, even a lateral protrusion in the ridges is possible to be observed (Figure 3B, second image). This local deformation of the ridges can also increase the contact area, and this should be addressed in further studies with a larger number of participants. One important limitation of the present study is the age range of participants from 26 to 55 years. For older individuals (>65 years), an increased keratinization and a breakdown of both the dermal–epidermal junction and the sensory corpuscles, such as MCs, has been reported [16,27]. Future investigations will address the morphology of the protrusions in individuals of high age.

The correlation between SC thickness and protrusion height, and the co-localization of protrusions with MCs suggests that the morphology of the glabrous skin is highly specialized for stimulus transmission. The SC, as a viscoelastic structure [28], is capable of deforming to reflect surface textures, where ridges offer an increase in surface contact. The micrometer-scale increase in the surface area at the protrusions may enhance the coupling of SC deformation to the receptor during tactile exploration. This idea could be tested by extending recent simulations of friction-induced strain at the top of papillae which contain MCs [24]. This peculiar local morphology of the epidermis and SC above MCs may thus have a function in the transmission of mechanical stimuli towards tactile perception, reminiscent of a remote control where the SC is the button with an increased area of contact and the MCs are the underlying electric switches (Figure 6). The local reduction in the epidermis thickness in the areas above MCs (Figure 4C) also indicates that MCs can project themselves into the SC, increasing their sensitivity. The observation of protrusion and thickness variation may be difficult in traditional histology with cross-sectional biopsies, since the tissue tends to stretch during material preparation and the protrusion scale is in the micrometer scale. Nevertheless, a careful look at classical skin histology results seems to also indicate the protrusion of the epidermis at the site of MCs, which were referred to by Cauna [29] as “elevations superficial to Meissner ‘s corpuscles” (Figures 10 and 11 in Ref. [29]).

In summary, our LSM and OCT data clarify conditions for the application of these in vivo non-invasive skin imaging techniques in studies of the morphology of the glabrous skin. We report the discovery of protrusions of the epidermis into the stratum corneum above the sites of Meissner corpuscles. The outstanding reflectance signal of these protrusions can guide LSM users to identify the location of Meissner corpuscles in the examination of glabrous skin with a stratum corneum, of which the thickness limits the depth range of the LSM. Taking the reflectance of the protrusion as proxy for the detection of a Meissner corpuscle may boost the efficiency of LSM as a method to determine the density of Meissner corpuscles in the study of diseases which affect tactile sensitivity. We suggest that a possible function of the reported protrusion lies in the morphological amplification of mechanical stimuli during tactile exploration of surface textures. This function needs further investigation by modeling skin mechanics and in situ imaging of stress-induced deformation of the stratum corneum. 

## 4. Materials and Methods

### 4.1. Study Design

Ten volunteers (3 males, 7 females) were enrolled in this study, with an age range from 23 to 55 years (average 34.7 years). The volunteers were instructed not to apply any cosmetic formulations, such as moisturizer lotions, to their hands for at least 24 h prior to the study. High-resolution images were obtained of the skin of the index and small fingers, as well as the tenar palm area. Optical coherence tomography (OCT) and laser scan microscopy (LSM) images were first acquired for one participant to understand the applications and limitations of each technique. After establishing the most suitable method to study MCs, OCT and LSM high-resolution images of all participants were recorded and analyzed with respect to the MC localization in glabrous skin.

### 4.2. Optical Coherence Tomography 

The OCT system (VivoSight Michelson Diagnostics Ltd., Maidstone, UK) was used in this study for in vivo evaluation of glabrous skin of the index fingers, small fingers, and the palm. This method detects the optical path length of light, which is backscattered from different biological tissues by low-coherence interferometry [30,31]. The reflected light interferes with a reference beam that stems from the same light source [32]. The VivoSight^®^ operates with a pulsed 1305 nm laser and a maximum repetition rate of 20 kHz, resulting in a lateral resolution of 7.5 µm and a penetration into the skin of up to 6 mm [30,33,34]. 

### 4.3. Laser Scan Microscopy 

The morphological characteristics of the different skin layers were evaluated by laser scan microscopy (VivaScope 1500 Multilaser, MAVIG GmbH, Munich, Germany), which uses a laser source with a wavelength of 488 nm or 785 nm, an immersion objective, and a camera for 20 images per second. The choice of 488 nm or 785 nm wavelengths depended on the preference for penetration depth or lateral resolution. The Vivastack imaging system provides multiple confocal images recorded at successive depths, typically with a size of 4.5-by-4.5 μm [35]. To acquire more detail, it is possible to change the image size in this device to 1.5-by-1.5 μm. The Vivascope is also capable of automatically recording and stitching images of a defined region of the skin in a horizontal plane [36]. The so-called VivaBlock^®^ images were acquired in an area of 3 mm × 3 mm, to study the distribution of receptors in the glabrous skin [15,18]. We measured all ten participants’ index fingers, small fingers, and tenar palm anatomic regions.

### 4.4. Statistics

Statistical analysis was performed using the software Graph Pad Prism 8.0. The data were tested for normality by the Shapiro–Wilk test and correlated using the ANOVA test to compare the three tested anatomic regions. For a comparison of two groups, the t-test was applied. Differences with *p* < 0.05 were denoted as significant.

## Figures and Tables

**Figure 1 ijms-24-07121-f001:**
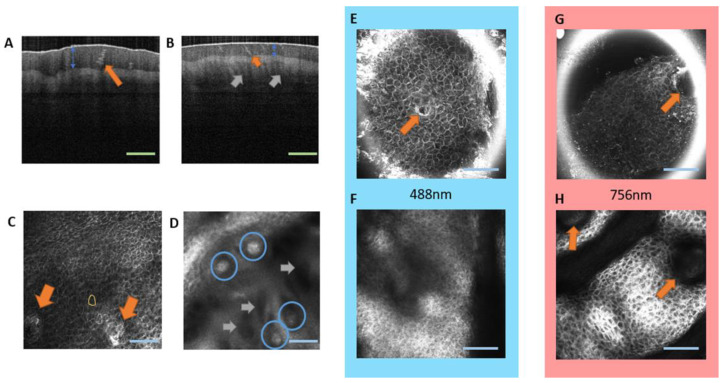
Comparison of imaging techniques for the glabrous skin. Longitudinal cross-sections recorded by OCT allow for the quantification of the SC thickness (blue arrows) for the index finger (**A**) and the small finger (**B**) of one participant, green scale bar 200 µm. Sweat glands (orange arrow) appear as helicoidal structures. Papillary structures are observed only in the small finger (**B**, gray arrows) due to the thinner SC. LSM images of the small finger recorded at 70 µm (**C**) and 240 µm (**D**) depth exhibit corneocytes (**C**, yellow contour) and the sweat glands (**C**, orange arrows), MCs at the DEJ (**D**, blue circles) as ellipses with high reflectance, and the papillae (**D**, gray arrows). LSM images recorded with the 488 nm laser have a better resolution for the SC at a depth of 20 to 70 µm (**E**) than the 756 nm laser (**G**), while the larger penetration depth of the 756 nm laser provides more detail in the epidermis and at the DEJ at a depth of 150 to 250 µm (**H**) compared to the 488 nm laser (**F**). Orange arrow indicate sweat glands, blue scale bar: 100 µm.

**Figure 2 ijms-24-07121-f002:**
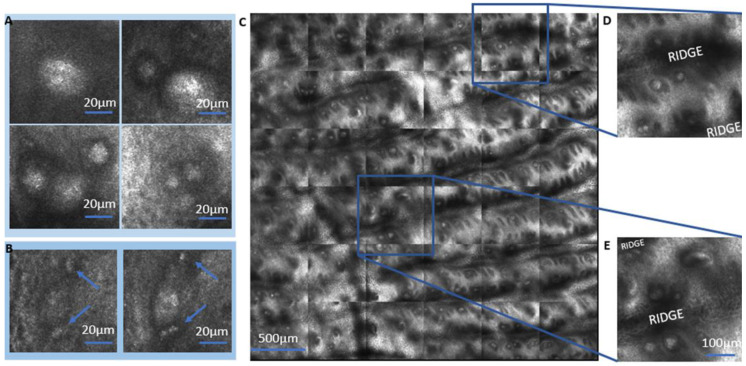
LSM observations of Meissner corpuscles. (**A**) The MCs can be single, in pairs or triplets in the same papilla occupying variable volume. (**B**) Proximity of MCs (center of the figure) and blood capillaries (marked with blue arrows) is often observed. (**C**) The VivaBlock^®^ image of a 9 mm² area demonstrates that MCs are most often found close to the ridges (**D**,**E**).

**Figure 3 ijms-24-07121-f003:**
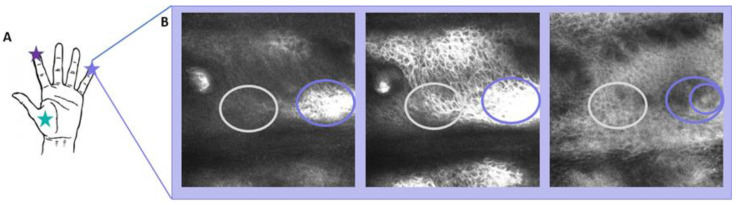
Meissner corpuscles create protrusions of the epidermis into the stratum corneum of glabrous skin. (**A**) Three anatomic positions were analyzed in this study (stars). (**B**) LSM images of a small finger recorded at a depth of 150, 159, and 210 µm. Regions containing an MC are marked by purple ellipses, equivalent regions with MC by gray ellipses.

**Figure 4 ijms-24-07121-f004:**
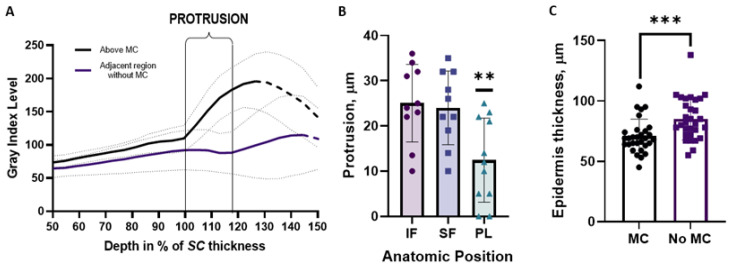
(**A**) Gray level index as a function of the depth as the percent of the stratum corneum thickness. Average data for the small finger of all participants are compared for regions with MCs and adjacent region without MCs. The onset of reflectance at a lower depth above the MC indicates a protrusion of the epidermis into the stratum corneum. The decay of GIL at a greater depth is due to the limited penetration depth of the light. The respective data points are represented by the dashed line. (**B**) Distribution of the protrusion height for index finger (IF), small finger (SF), and tenar palmar region (PL). Protrusions are higher in the finger skin compared to the palm (** *p* = 0.003). (**C**) The epidermis without MCs is thinner for the regions with MCs (*** *p* < 0.001).

**Figure 5 ijms-24-07121-f005:**
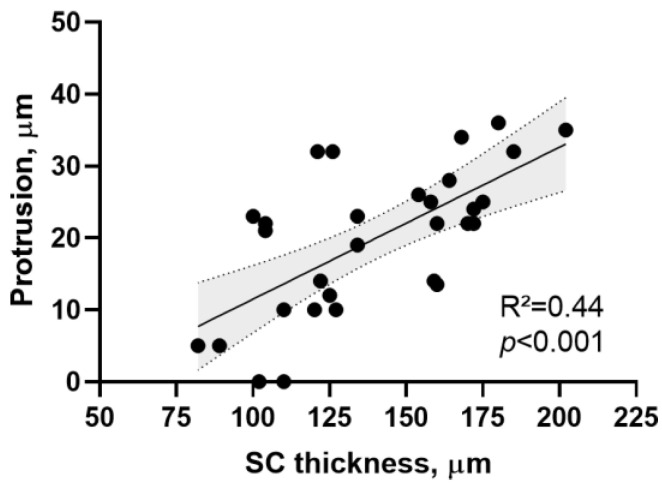
Positive correlation between SC thickness and the height of protrusions (R^2^ = 0.49; *p* < 0.001).

**Figure 6 ijms-24-07121-f006:**
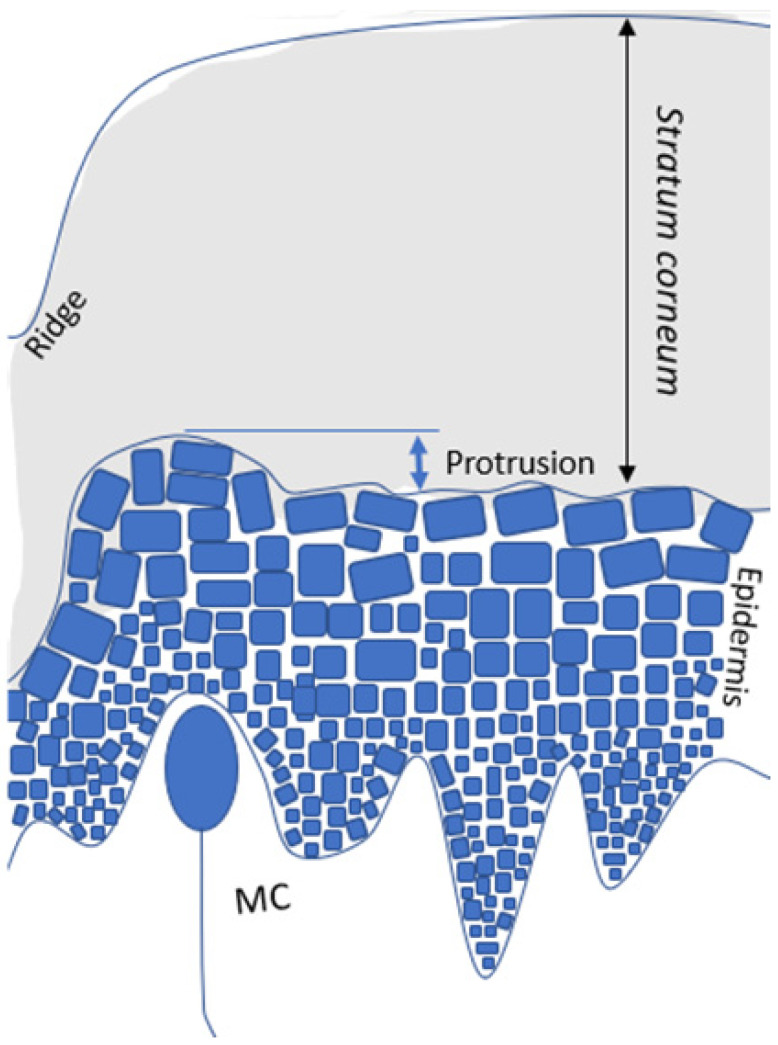
Schematic representation of results. In glabrous skin, Meissner corpuscles (MCs) are co-localized with protrusions of the epidermis into the stratum corneum and with a local reduction in the epidermis thickness.

**Table 1 ijms-24-07121-t001:** Morphological information regarding the participants in this study. Above MCs, the thickness of the epidermis was reduced, and the epidermis protruded into the stratum corneum.

Participant/Age/Sex	Anatomic Position	SC Thickness (µm)	Epidermis Thickness (µm) MC/No MC	Protrusions (µm)
001/32/F	Index finger	180	72/86.5	23
	Small finger	126	55/55	33
	Palm	107	59/67	23
002/26/F	Index finger	134	74/85	13.5
	Index finger	128	70/81	10
	Small finger	102	66/72	19
	Small finger	102	60/69	31
003/30/M	Index finger	190	95/103	32
	Small finger	157	112/138	14
	Palm	89	45/76	14
004/29/F	Index finger	184	93/105	37
	Index finger	180	76/96	22
	Small finger	164	59/67	10
	Palm	112	63/102	0
005/29/F	Index finger	139	70/103	25
	Index finger	136	60/89	24
	Palm	88	54/66	5
006/55/F	Small finger	170	75/83	22
	Palm	125	59/66	12
007/52/F	Small finger	127	65/59	26
	Palm	100	65/92	0
008/44/M	Index finger	190	72/88	35
	Palm	110	64/81	10
	Palm	92	64/77	5
009/32/F	Index finger	168	75/85	31
	Small finger	121	65/63	22
	Small finger	113	62/55	31
	Palm	104	59/58	22
010/23/M	Small finger	202	132/115	28
	Palm	158	85/65	25
	Palm	156	88/71	21

## Data Availability

The data that support the findings of this study are available on request from the corresponding author. The data are not publicly available due to privacy or ethical restrictions.

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
