# Peer review of "Revealing the Meissner Corpuscles in Human Glabrous Skin Using In Vivo Non-Invasive Imaging Techniques"

_ijms, 2023, doi:10.3390/ijms24087121_

Round 1
Reviewer 1 Report
Authors aim was to use Optical Coherence Tomography (OCT) and Laser Scan Microscopy (LSM), new non-invasive optical microscopy techniques, as a non‐invasive diagnostic tool to study and count the Meissner corpuscles present in the skin of index and small fingers as well as the tenar palm area of patients affected by tactile sensitivity. In this study, OCT did not provide with images of Meissner corpuscles, while LSM allowed to identify their location since the regions containing these corpuscles could be easily identified by an enhanced optical reflectance above them due to protrusions of the epidermis into the stratum corneum above the corpuscles’ sites.
Images are of good quality; the text is well written. Limitation: This diagnostic tool can be applied only in some specialized hospitals.
As Minor comments pay attention on some misprints such as: Pag.2, row 91 …..laser is provides….delete is
Author Response
#1
Authors aim was to use Optical Coherence Tomography (OCT) and Laser Scan Microscopy (LSM), new non-invasive optical microscopy techniques, as a non‐invasive diagnostic tool to study and count the Meissner corpuscles present in the skin of index and small fingers as well as the tenar palm area of patients affected by tactile sensitivity. In this study, OCT did not provide with images of Meissner corpuscles, while LSM allowed to identify their location since the regions containing these corpuscles could be easily identified by an enhanced optical reflectance above them due to protrusions of the epidermis into the stratum corneum above the corpuscles’ sites.
Images are of good quality; the text is well written. Limitation: This diagnostic tool can be applied only in some specialized hospitals.
Thank you for your review. We agree that the distribution of the diagnostic device is limited but for such investigation indispensable because the high resolution and noninvasive application. We have added the following phrase in the lines 189-190:
At present, the availability of LSM is limited, but indispensable for such studies if biopsies will be avoided.
As Minor comments pay attention on some misprints such as: Pag.2, row 91 …..laser is provides….delete is
Thank you for your review and suggestion. The following consideration was corrected in the paper and other minor errors were reviewed and corrected.

Reviewer 2 Report
in the future I think it will be valuable to do comparative study regarding the accuracy between the used noninvasive devices and electron microscopy
Author Response
#2
in the future I think it will be valuable to do comparative study regarding the accuracy between the used noninvasive devices and electron microscopy.
Thank you for your review, we have included this valuable suggestion for our future study.

Reviewer 3 Report
The study entitled "Revealing the Meissner Corpuscles in human glabrous skin using in vivo non-invasive imaging techniques" is of great interest since it proposes a non-invasive way to replace the traditional skin biopsy used up to now by imaging techniques, which could be of special application in multiple pathologies that affect the peripheral nervous system, such as peripheral neuropathies, and that present with healing difficulties, for which reason invasive techniques are often ruled out.
From my point of view, different appreciations should be taken into account:
- It would be a good justification to add this small point of healing problems in patients with peripheral pathologies for the use of the proposed imaging techniques.
- It is recommended to put the initials SC for stratum corneum on line 39, since the initials are used later but they do not appear in the text at any point.
- Some authors consider that Merkel cells are located in the DEJ and others in the basal layer of the epidermis; they must take this information into account to make the clarification (line 43).
- Lines 49 and 55: it is recommended to add a current bibliographic reference that exists in humans (see García-Mesa et al., 2021), since skin biopsy is used for the study of patients with painful and painless diabetic neuropathy, and it is a current study.
- In Figure 1, the structures that are referenced with the letters e-h are not identified. It is recommended to add some "mark" to identify structures and to be able to compare image resolution.
- In Figure 2: several questions arise that must be resolved: What error is estimated in the identification of the Meissner corpuscles since the intensity is similar to that of the capillaries? Is the presence of sweat gland ducts in the dermal papillae taken into account? From the point of view of a morphologist, there is not enough clarity for the identification of these structures as Meissner corpuscles if the previous questions are not resolved.
- In Table 1: the age of the participants is between 26-55 years, but the experience of our research group in cutaneous sensory corpuscles has allowed us to previously demonstrate that, at older ages (>65 years), there is a increased keratinization in certain skin areas and a breakdown of both the DEJ and the sensory corpuscles. Do you think these biases could be avoided with age or could it only be used in young subjects? If these biases are not contemplated, how can it be given clinical applicability in peripheral neuropathies or other diseases that present with alterations in the peripheral nervous system when there is a loss of structure of the different skin layers, a "strangulation" of the papillae skin, a thickening of the SC and a loss of morphology of the sensitive corpuscles? These questions must be raised.
- In Figure 4, graph A that measures the Gray Index Level indicates that the area with the most protrusions is not the one that contains the highest Meissner corpuscle index. Clarification is requested, in case the interpretation is not correct.
Figure 6 is very representative for the understanding of the discovery carried out by means of imaging techniques, but, in paraffin-embedded preparations on which IHC techniques are performed, these protrusions have not been observed in relation to the dermal papillae that present corpuscles of Meissner, but rather that they are observed in the CS of those groups of dermal papillae that have a greater vertical axis. How can you explain that your results are not biased by imaging techniques or by not including a parameter such as the one mentioned above?
From my point of view, it is a well-planned study but with certain doubts in the data provided since there are biases that have not been taken into account. It must be reviewed for acceptance. Thank you so much.
Author Response
#3
The study entitled "Revealing the Meissner Corpuscles in human glabrous skin using in vivo non-invasive imaging techniques" is of great interest since it proposes a non-invasive way to replace the traditional skin biopsy used up to now by imaging techniques, which could be of special application in multiple pathologies that affect the peripheral nervous system, such as peripheral neuropathies, and that present with healing difficulties, for which reason invasive techniques are often ruled out.
Thank you for your valuable review and suggestions. We have revised and incorporated into the manuscript as maximum as we could according to your comments and suggestions.
From my point of view, different appreciations should be taken into account:
- It would be a good justification to add this small point of healing problems in patients with peripheral pathologies for the use of the proposed imaging techniques.
We have included the following statement and the reference (García-Mesa et al., 2021) indicated by the reviewer in lines 54-56:
Variations in the amount, physiology, and distribution of MCs can potentially serve as markers for disorders or patients with healing problems from diabetic neuropathy, as described by Garcia-Mesa et al [16].
- It is recommended to put the initials SC for stratum corneum on line 39, since the initials are used later but they do not appear in the text at any point.
We have corrected this.
- Some authors consider that Merkel cells are located in the DEJ and others in the basal layer of the epidermis; they must take this information into account to make the clarification (line 43).
We have changed the paragraph to: “These nerves lead to the perception of subtle tactile details and are localized in the basal layer of the epidermis (Merkel cells), the dermal-epidermal junction (DEJ) (Merkel cells, Meissner Corpuscles) or the dermis (Pacinian corpuscles and Ruffini organs).”
- Lines 49 and 55: it is recommended to add a current bibliographic reference that exists in humans (see García-Mesa et al., 2021), since skin biopsy is used for the study of patients with painful and painless diabetic neuropathy, and it is a current study.
We have included the recommended literature in line 56 and added the following statement: “However, biopsies are an invasive and uncomfortable method to quantify mechanoreceptors in healthy participants in fundamental physiology studies.”
- In Figure 1, the structures that are referenced with the letters e-h are not identified. It is recommended to add some "mark" to identify structures and to be able to compare image resolution.
Thank you for your consideration. We have included in Figure 1 some arrows in the pointed images to show the presence of sweat glands.
- In Figure 2: several questions arise that must be resolved: What error is estimated in the identification of the Meissner corpuscles since the intensity is similar to that of the capillaries? Is the presence of sweat gland ducts in the dermal papillae taken into account? From the point of view of a morphologist, there is not enough clarity for the identification of these structures as Meissner corpuscles if the previous questions are not resolved.
The application of LSM in the identification of Meissner Corpuscles is already described in the literature. For example, in the study of Creigh et al. 2022 (1), the identification of Meissner Corpuscles is presented in figure 4. Other bibliographic references were also important in the delimitation regarding Meissner Corpuscles observation using LSM (2-4). Hermann et al (2007) (2) had performed a study comparing biopsies with LSM images to confirm the morphology of these structures. These previous studies helped us to clarify how to identify Meissner Corpuscles using the present technique. We have also utilized biopsies studies to understand the morphological aspects of Meissner Corpuscles and to apply in studies using LSM (5,6). Based on these previous studies, we were able to identify the Meissner Corpuscles in the LSM images.
The question regarding the sweat glands is very important to be addressed since a non-trained evaluator could confuse both in the dermal-epidermal junction. However unmistakable differences are observed:
- Meissner Corpuscles are located close to the ridges while sweat glands are located more towards the central part of the ridges.
- Meissner Corpuscles appear with higher reflectance and present an ellipse shape, which distinguishes then from other structures.
We have added one figure to the Supplementary Material clarify the distinction of sweat glands (blue arrows) from Meissner corpuscles (orange arrows) and to further visualize the presence of protrusions (orange ellipses). In the text, the following phrase was included: “The appearance of MCs and their distinction from sweat glands is further demonstrated in Sup. Fig.1 in the Supplementary Material.”
Using LSM, it is possible to observe the flow of blood cells in vivo during the evaluation. As pointed out by Greenwold et al. (2009), using LSM it is possible to even record movies of the blood flow. In this manuscript, we do not present such movies of the blood flow, but as mentioned in the text, using stacks of images at a height difference of 1.5µm, is possible to observe this flow. To further demonstrate the identification of Meissner Corpuscles, we have included supplementary material that is attached here. The orange ellipse indicates the protrusion, the orange arrows point to the MCs, and the green arrow follows the blood flow which is observed in the sequence of images.
We included the following statement in the paper lines 107-110: “The Supplementary Material provides additional images of MC morphologies and of blood capillaries in the dermis (Sup. Fig. 2). For a better representation of the blood capillaries in the dermal-epidermal junction, VivaStack images should be recorded with 1.5µm height increment.”
- In Table 1: the age of the participants is between 26-55 years, but the experience of our research group in cutaneous sensory corpuscles has allowed us to previously demonstrate that, at older ages (>65 years), there is an increased keratinization in certain skin areas and a breakdown of both the DEJ and the sensory corpuscles. Do you think these biases could be avoided with age or could it only be used in young subjects? If these biases are not contemplated, how can it be given clinical applicability in peripheral neuropathies or other diseases that present with alterations in the peripheral nervous system when there is a loss of structure of the different skin layers, a "strangulation" of the papillae skin, a thickening of the SC and a loss of morphology of the sensitive corpuscles? These questions must be raised.
This is an interesting suggestion for future work and we appreciate that the reviewer shares the experience of his or her research group. We discuss the possible bias arising from the limited age range in this initial study by including the following statement in lines 216-220: “One important limitation of the present study is the age range of participants from 26 to 55 years. For older individuals (>65 years), an increased keratinization and a breakdown of both the dermal-epidermal junction and the sensory corpuscles such as MCs has been reported [16,27]. Future investigations will address the morphology of the protrusions in individuals of high age.”
- In Figure 4, graph A that measures the Gray Index Level indicates that the area with the most protrusions is not the one that contains the highest Meissner corpuscle index. Clarification is requested, in case the interpretation is not correct.
The interpretation of the reviewer is indeed incorrect. The graph A does not give any information about the number of protrusions. It gives information about the average GIL as function of the relative depth in percentage of the SC thickness. We have clarified by changing the legend to “above MC” and “adjacent area” to establish a better connection to the text. To avoid any additional misunderstood we focused the graphic in the protrusion area. An explanatory sentence has been added:
“One example is given in Fig. 4A, where the average intensity level is plotted as function of the depth for the small finger of all participants. Note that the depth is taken as percentage of the thickness of each participant's stratum corneum. Above MCs, the reflectance increases at the top of the protrusion in a depth which corresponds to 100% of the SC. In adjacent regions without MCs, the reflectance increases only at the dermal-epidermal junction in a depth corresponding to 118% of the stratum corneum.” Lines 144-149.
Figure 6 is very representative for the understanding of the discovery carried out by means of imaging techniques, but, in paraffin-embedded preparations on which IHC techniques are performed, these protrusions have not been observed in relation to the dermal papillae that present corpuscles of Meissner, but rather that they are observed in the CS of those groups of dermal papillae that have a greater vertical axis. How can you explain that your results are not biased by imaging techniques or by not including a parameter such as the one mentioned above?
It is interesting to discuss strengths and limitations of each technique. Biopsies are very useful and precise to provide details of the morphology and histology, but when a histological cut is performed, some structural alterations can happen. As pointed out by Hermann et al (2007) (2) “comparison of MC densities, using estimates from in vivo RCM, and estimates from fixed biopsy tissue, is complex because skin biopsies undergo fixation shrinkage, which is difficult to precisely correct for”. Along this line or argument, the observation of protrusions may in samples from biopsies may be distorted by fixation shrinkage, for example. LSM can provide images in vivo, avoiding this limitation. We believe that our manuscript provides enough detail for the reader to understand and evaluate our imaging techniques.
We have addressed the concerns of the reviewer in the lines 232-236: “The observation of protrusion and thickness variation may be difficult in traditional histology with cross-sectional biopsies since the tissue tends to stretch during material preparation and since the protrusions scale is in the micrometer scale. Nevertheless, a careful look at classical skin histology results seems to also indicate the protrusion of the epidermis at the site of MCs, which were referred to by Cauna [29] as “elevations superficial to Meissner ‘s corpuscles” (Figs. 10 and 11 in Ref. 29).” The findings of Cauna (1954) (5) support our observation of superficial elevations above MCs using biopsy.
From my point of view, it is a well-planned study but with certain doubts in the data provided since there are biases that have not been taken into account. It must be reviewed for acceptance. Thank you so much.
Thank you very much for your valuable remarks on our work. We hope that the present version with all the corrections is now suitable for publication.
References:
(1) Creigh, Peter D., et al. "In Vivo Reflectance Microscopy of Meissner Corpuscles and Bedside Measures of Large Fiber Sensory Function: A Normative Data Cohort." Neurology 98.7 (2022): e750-e758.
(2) Herrmann, David N., et al. "In vivo confocal microscopy of Meissner corpuscles as a measure of sensory neuropathy." Neurology 69.23 (2007): 2121-2127.
(3) Creigh, Peter D., et al. "In-vivo reflectance confocal microscopy of Meissner's corpuscles in diabetic distal symmetric polyneuropathy." Journal of the neurological sciences 378 (2017): 213-219.
(4) Almodovar, Jorge L., et al. "HIV neuropathy: an in vivo confocal microscopic study." Journal of neurovirology 18.6 (2012): 503-510.
(5) Cauna, N. Nature and functions of the papillary ridges of the digital skin. Anat Record, 1954, 119(4), 449-468.
(6) A Vega, Jose, et al. "Clinical implication of Meissner's corpuscles." CNS & Neurological Disorders-Drug Targets (Formerly Current Drug Targets-CNS & Neurological Disorders) 11.7 (2012): 856-868.

Round 2
Reviewer 1 Report
authors correctly replied to the referee' questions
Author Response
Thank you very much for your review.